

# Association between Dietary Inflammatory Index and Type 2 diabetes mellitus in Xinjiang Uyghur autonomous region, China

WenHui Fu[1,2], Hualian Pei[1], Nitin Shivappa[3,4], James R. Hebert[3,4], Tao Luo[1], Tian Tian[1], Dilibaier Alimu[1], Zewen Zhang[1] and Jianghong Dai[1]

[1] Department of Epidemiology and Biostatistics, School of Public Health, Xinjiang Medical University, Urumqi, Xinjiang Uygur Autonomous Region, China
[2] Department of Immunization Programme, Xinjiang Uygur Autonomous Region Center for Disease Control and Prevention, Urumqi, Xinjiang Uygur Autonomous Region, China
[3] Department of Epidemiology and Biostatistics, Arnold School of Public Health, University of South Carolina, Columbia, SC, United States of America
[4] Cancer Prevention and Control Program, University of South Carolina, Columbia, SC, United States of America

## ABSTRACT

**Background**. Diet and inflammation have both been studied in relation to type 2 diabetes mellitus (T2DM). The aim of this cross-sectional study was to examine the association between the Dietary Inflammatory Index (DII®) and T2DM.

**Methods**. Subjects were adults enrolled in the baseline study of the Xinjiang multi-ethnic natural population cohort and health follow-up study from January to May 2019. The study involved 5,105 subjects (58.7% men) between 35 and 74 years of age. The DII score was calculated from a data obtained via a food frequency questionnaire consisting of 127 food items.

**Results**. Logistic regression analyses were used to estimate the odds ratios (ORs) and 95% confidence intervals (95% CIs) of DII in relation to T2DM. After adjusting for potential confounders, compared to subjects in the 1st DII quintile, subjects in the 5th quintile (i.e., with the most pro-inflammatory diet) had higher odds of T2DM (OR = 3.27, 95%CI:2.38,4.50; $p < 0.001$).

**Conclusions**. Our results suggest that a pro-inflammatory diet is associated with a higher risk of T2DM in this population of Chinese adults.

## INTRODUCTION

Type 2 diabetes mellitus (T2DM) is a chronic metabolic disease (*Zuo, Shi & Hussain, 2014*). Current projections estimate that between 2017 and 2045 the number of people suffering from diabetes will increase more than 50%, with the vast majority diagnosed with T2DM (*Cho et al., 2018*). In 2017, the prevalence of diabetes in China was 11.2% (*Wang et al., 2020*).

Corresponding author
Jianghong Dai, epidjh@163.com

Of the many metabolic and lifestyle factors that are known to affect the development and complications of T2DM, inflammation has been gaining attention for its role in T2DM pathogenesis (*Dregan et al., 2014*). Overexpression of proinflammatory cytokines, such as Interleukin-1$\beta$ (IL-1$\beta$), Interleukin-6 (IL-6), and Tumor necrosis factor-$\alpha$ (TNF-$\alpha$), induce chronic inflammation in T2DM (*John, 2004*; *Mahlangu et al., 2019*; *Pradhan et al., 2001*), although the specific mechanism remains unclear. There are both modifiable and unmodifiable factors associated with inflammation; dietary factors are potentially modifiable and, therefore, have received a great deal of attention (*Moazzen et al., 2020*).

The effect of diet on inflammation can be assessed based on food-group, nutrition, or analysis of dietary patterns (*Barbaresko et al., 2013*). Because humans consume many nutrients through a variety of foods, a person's inflammatory status can be influenced by the balance between nutrients and other dietary components that promote or inhibit inflammation (*Minihane et al., 2015*). Therefore, in order to illustrate the complex effects of dietary components on inflammatory status, a better assessment method is needed to evaluate the impact of the overall diet on individual inflammatory status.

The Dietary Inflammatory Index (DII®) has been developed to characterize the inflammatory potential of the diet (*Cavicchia et al., 2009*). The design of this index included careful review and scoring of results from peer-reviewed research studies from human populations, qualifying cell culture experiments, and animal experiments (*Shivappa et al., 2014*). The validity of the DII has been demonstrated using different inflammatory biomarkers (*Kotemori et al., 2020*; *Tabung et al., 2015*). In addition, it is associated with components of metabolic syndrome (*Ferreira et al., 2019*) and a wide variety of chronic diseases (*Phillips et al., 2019*).

Xinjiang is a multi-ethnic region where the Han, Uygur, and Kazak groups account for more than 90% of the total population. Because of differences in customs and living environments, there is a certain heterogeneity in dietary structure among these different ethnic groups. According to the physical examination data for all citizens in Xinjiang in 2018, the detection rate of T2DM varied among different populations. Thus, the aim of this study was to explore the relationship between the DII and T2DM in different participants in the Xinjiang multi-ethnic natural population.

## MATERIALS & METHODS

### Study population

The Xinjiang multi-ethnic natural population cohort construction and health follow-up study is a population-based prospective study designed by our group and implemented from January to May 2019. Participants in this study were members of a subsample in the baseline study, aged 35–74 years, and living in Yili Prefecture, Xinjiang Province in China.

Potential participants were invited if they met the following criteria for inclusion: (1) participants of the original cohort; (2) provided basic information, including diet and physical activity; (3) provided stored serum samples and blood analysis data. A total of 8367 adults, aged 35–74 years were enrolled. For the present analysis, we excluded participants with no fasting glucose data ($n = 2166$), then we excluded participants with

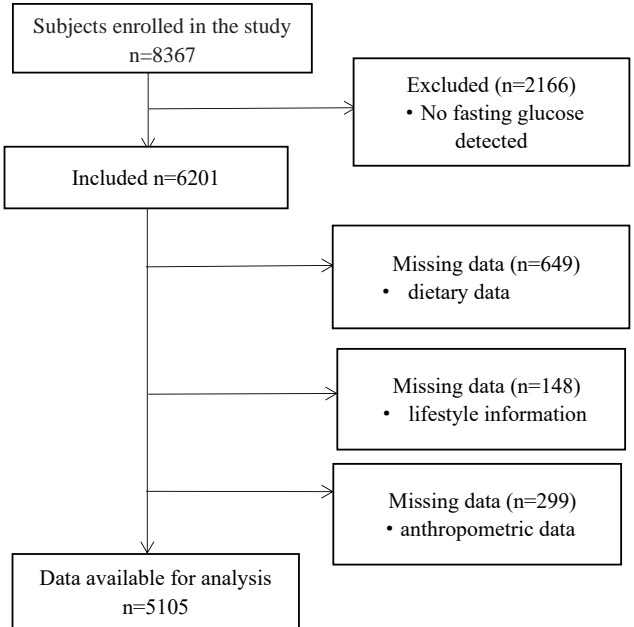

**Figure 1  Flow diagram of subjects included in the study.**

missing information on important covariates ($n = 797$) or with incomplete anthropometric data ($n = 299$). Finally, a total of 5,105 subjects were included in our study (Fig. 1).

All recruited subjects gave written informed consent for inclusion before they participated in the study. The study was conducted in accordance with the Declaration of Helsinki, and the protocol was approved by the Ethics Committee of Ethics Committee of Xinjiang Uygur Autonomous Region Academy of Traditional Chinese Medicine (2018XE0108).

## Dietary assessment

Habitual diet over the past year was assessed using the interviewer-administered food frequency questionnaire (FFQ) (*Hailili et al., 2020*). Referring to the dietary questionnaire of Xinjiang multi-ethnic natural population cohort construction and health follow-up study and combining the characteristics of a typical Xinjiang diet, the final questionnaire included 127 foods items. Consumption frequency of each food was queried on a daily, weekly, or monthly basis and converted into daily intake. The FFQs were collected by trained personnel.

## DII Score

Dietary data from the FFQs were used to calculate DII scores for each participant. The design and development of the DII has been described previously. Briefly, DII scores were calculated using a scoring algorithm, which was based on a review of 1,943 articles between 1950 and 2010 that reported on 45 different food parameters that were found to be associated with six inflammatory biomarkers (IL-1$\beta$, IL-4, IL-6, IL-10, and C-reactive

protein). Each study was assigned a "food parameter-specific inflammatory effect score". If the food parameter was pro-inflammatory, it was assigned a +1. If it was anti-inflammatory, it was assigned a −1. If the parameter did not change significant in terms of inflammation biomarkers, they were assigned a zero. The specific calculation steps were as follows: (1) dietary intake data were collected from study subjects using the FFQ; (2) dietary intake data were transformed to the food ingredients contained in the DII scale, according to the latest China Food Composition Table; (3) dietary intake data were compared with the global standard dietary intake database and a Z-score was calculated for each nutrient in the study according to the mean and standard deviation of each nutrient intake; (4) Z-scores were converted to centered proportions; (5) food parameter specific centered proportion is multiplied by the specific inflammatory effect score of each nutrient to derive food parameter specific DII score; (5) DII scores were then summed across over all food parameters to obtain an individual's specific DII score.

In our study, out of 45 possible foods, individual DII scores were calculated using 27 foods and nutrients for which we obtained intake data from FFQ. The 27 foods and nutrients included carbohydrate, protein, fat, alcohol, fiber, cholesterol, saturated fatty acid, mono-unsaturated fatty acid, poly-unsaturated fatty acid, omega 3 fat, omega 6 fat, trans fat, niacin, thiamin, riboflavin, vitamin B12, vitamin B6, iron, magnesium, zinc, vitamin A, vitamin C, vitamin E, folic acid, beta carotene, garlic, and onion.

## Biomarker evaluation

Fasting venous blood samples were collected from the anterior cubital vein of each participant (fasting time $\geq 8$ h). Small portions of serum were stored in refrigerated bottles and transported to laboratories, where they were then stored in equal portions at $-80\ ^\circ$C for further analysis.

## Definition of T2DM

For the present study, participants who had previously been diagnosed with diabetes at the county hospital level or above were labeled as "previously diagnosed". Subjects with fasting glucose concentration $\geq 7.0$ mmol/L during the study period were defined as having fasting glucose and/or HbA1c values that met the diagnostic criteria for T2DM and were also defined as T2DM patients.

## Anthropometric assessment

Weight and height were measured while wearing only light clothing, no shoes, and using standard scales. Body mass index (BMI) was calculated as weight/height$^2$ (kg/m$^2$) and classified according to the criteria of the Blue Book on Obesity Prevention and Control in China: underweight ($<18.5$ kg/m$^2$), normal weight ($18.5–23.9$ kg/m$^2$), overweight ($24–27.9$ kg/m$^2$), or obese ($\geq 28$ kg/m$^2$).

## Other participant characteristics

Investigators used a standardized questionnaire to collect information regarding the participants' demographic characteristics (*Hailili et al., 2020*) (e.g., age, sex, education, marital status), self-perception of body weight, past medical history, current medication

use, lifestyle information (e.g., physical activity, smoking status, alcohol consumption, etc.) and information on reproductive history (for females).

## Statistical analysis

We performed a descriptive analysis of the main characteristics of interest to assess compliance with the model's assumptions. We analyzed the characteristics difference between diabetic and non-diabetic subjects. $P$-values were determined using a student's $t$-test for continuous variables and Chi-square test for categorical variables. Moreover, Kolmogorov–Smirnov test was used to check the normality. Categorical variables were presented as frequencies of occurrence and percentages, and continuous variables were presented as the mean and standard deviation (SD). One-way analysis of variance (ANOVA) was used to examine whether there were differences across DII quintiles, while the chi-square test was used to assess the distribution of qualitative variables over DII quintiles.

To evaluate the magnitude of the association between the DII and diabetes, we used a simple univariable (unadjusted) and multivariable logistic regression analysis adjusted for baseline characteristics (sex, age, ethnicity, physical activity, smoking, education, alcohol intake, body mass index). In order to asses possible effect modification, analyses stratified by physical activity, BMI, ethnic and age were performed. $P$-$_{\text{Interaction}}$ was calculated using the multiplicative interaction term. We tested the significance using a likelihood ratio test by comparing a model with the main effects of each intake and the stratified variable and the reduced model interaction terms with only the main effect. All $p$-values presented are two-tailed; $p < 0.05$ was considered significant.

## Sensitivity analysis

Several sensitivity analyses were performed. First, we adjusted different models when multiple imputations were applied for all covariates. Second, we calculated the E-values to appraise unmeasured confounding. E-values were calculated by the online $E$-value calculator (https://mmathur.shinyapps.io/evalue/).

All statistical analyses were performed using SPSS 21.0 software and SAS® 9.4 (SAS Institute Inc., Cary, NC). All $p$ values were based on two-sided tests.

## RESULTS

The baseline characteristics of participants are shown in Table 1. A total of 5,105 subjects (58.7% male) were included in the present analysis. The main age of the participants ranged from 45 to 60 years, the mean DII of all participants was 0.81, and the overall prevalence of T2DM was 8.2%. Furthermore, the mean DII was significant higher in participants with T2DM ($p < 0.001$).

According to DII score quintiles, the participants with the most pro-inflammatory diets were significant older ($p = 0.004$) and more likely to be obese ($p = 0.03$). These participants also had lower educational status, lower physical activity level, and higher blood glucose. The distribution of DII by ethnicity showed a higher proportion of ethnic minorities with pro-inflammatory diets (Table 2).

Whether using DII as a continuous or categorical variable, the correlation between T2DM and DII was evident. After adjusting for ethnicity and sex, the odds of having

**Table 1  Characteristics of the study population.**

| | Overall study (n = 5105) | Non-T2DM subjects (n = 4686) | T2DM subjects (n = 419) | p-value |
|---|---|---|---|---|
| Sex, n (%) | | | | |
| Male | 2998(58.7) | 2769(59.1) | 229(54.7) | 0.08 |
| Female | 2107(41.3) | 1917(40.9) | 190(45.4) | |
| Age groups, n (%) | | | | <0.001 |
| <45 | 1662(32.6) | 1605(34.3) | 57(13.6) | |
| 45–60 | 2323(45.5) | 2132(45.5) | 191(45.6) | |
| >60 | 1120(22.0) | 949(20.3) | 171(40.8) | |
| Weight status, n (%) | | | | 0.60 |
| BMI<24 kg/m$^2$ | 1584(31.0) | 1462(31.2) | 122(29.1) | |
| BMI 24–28 kg/m$^2$ | 1837(36.0) | 1678(35.8) | 164(37.9) | |
| BMI>28 kg/m$^2$ | 1684(33.0) | 1546(33.0) | 138(32.9) | |
| Ethnicity, n (%) | | | | |
| Han | 665(13.0) | 590(12.6) | 75(17.9) | <0.001 |
| Kazak | 919(18.0) | 879(18.8) | 40(9.6) | |
| Hui | 1302(25.5) | 1153(24.6) | 149(35.6) | |
| Uyghur | 746(14.6) | 661(14.1) | 85(20.3) | |
| Others | 1473(28.9) | 1403(29.9) | 70(16.7) | |
| Educational level, n (%) | | | | |
| Low | 3695(73.1) | 3378(72.8) | 317(73.1) | 0.34 |
| Medium | 1337(26.4) | 1237(26.7) | 100(23.9) | |
| High | 25(0.5) | 24(0.5) | 1(0.2) | |
| Smoking status, n (%) | | | | |
| Never | 3980(78.0) | 3643(77.7) | 337(80.4) | 0.21 |
| Once in a while | 106(2.1) | 95(2.0) | 11(2.6) | |
| Usually | 1019(20.0) | 948(20.0) | 71(17.0) | |
| Alcohol status, n (%) | | | | |
| Never | 4300(84.2) | 3933(83.9) | 367(87.5) | 0.05 |
| Once in a while | 686(13.4) | 645(13.8) | 41(9.8) | |
| Usually | 119(2.3) | 108(2.3) | 11(2.6) | |
| Physical activity level, n (%) | | | | |
| Low | 2862(56.1) | 2571(54.9) | 291(69.5) | <0.001 |
| Medium | 2106(41.3) | 1983(42.3) | 123(29.4) | |
| High | 137(2.7) | 132(2.8) | 5(1.2) | |
| Glucose (mmol/L) | 5.12(0.02) | 4.85(0.01) | 8.10(0.2) | <0.001 |
| DII | 0.81(0.1) | 0.76(0.1) | 1.44(0.2) | <0.001 |

T2DM across all DII score quintiles were 1.00, 1.12, 1.17, 1.18, and 3.37 (95% CI: 2.46, 4.63; $p < 0.001$). Finally, after additional adjustment for variables, we observed that subjects with the most pro-inflammatory diets had higher odds of having T2DM (OR: 3.27, 95%

**Table 2** Baseline characteristics of study population by Dietary Inflammatory Index (DII) categorization.

| | DII quintiles | | | | | p-value |
|---|---|---|---|---|---|---|
| | Q1:most anti-inflammatory | Q2 | Q3 | Q4 | Q5:most pro-inflammatory | |
| Median (range) | −5.3 (≤−3.0) | −1.3 (−2.9–0.04) | 1.3 (0.05–2.4) | 3.6 (2.5–4.4) | 5.8 ≥4.5 | <0.001 |
| Sex, n (%) | | | | | | 0.19 |
| Men | 618(61.4) | 574(56.7) | 590(59.2) | 573(57.0) | 643(59.2) | |
| Women | 388(38.6) | 438(43.3) | 406(40.8) | 432(43.0) | 443(40.8) | |
| Age groups, n (%) | | | | | | 0.004 |
| <45 | 368(36.6) | 321(31.7) | 322(32.3) | 327(32.5) | 324(29.8) | |
| 45–60 | 452(44.9) | 483(47.7) | 448(45.0) | 457(45.5) | 483(44.5) | |
| >60 | 186(18.5) | 208(20.6) | 226(22.7) | 221(22.0) | 279(25.7) | |
| Weight status, n (%) | | | | | | 0.03 |
| BMI<24 | 290(28.8) | 308(30.4) | 310(31.1) | 331(32.9) | 345(31.8) | |
| BMI 24–28 | 386(38.4) | 369(36.5) | 358(35.9) | 377(37.5) | 347(32.0) | |
| BMI>28 | 330(32.8) | 335(33.1) | 328(32.9) | 297(29.6) | 394(36.3) | |
| Ethnicity, n (%) | | | | | | <0.001 |
| Han | 166(16.5) | 162(16.0) | 131(13.2) | 116(11.5) | 90(8.30) | |
| Kazak | 127(12.6) | 182(18.0) | 177 (17.8) | 163(16.2) | 270(24.9) | |
| Hui | 304(30.2) | 244(24.1) | 262(26.3) | 249(24.8) | 243 (22.4) | |
| Uyghurs | 127(12.6) | 132(13.0) | 141(14.2) | 179(17.8) | 167(15.4) | |
| Others | 282(28.0) | 292(28.9) | 285(28.6) | 298(29.7) | 316(29.1) | |
| Educational level, n (%) | | | | | | <0.001 |
| Low | 672(67.7) | 718(71.4) | 722(73.1) | 759(76.1) | 824(76.8) | |
| Medium | 316(31.8) | 283(28.1) | 259(26.2) | 235(23.6) | 244(22.7) | |
| High | 5(0.5) | 5(0.5) | 7(0.7) | 3(0.3) | 5(0.5) | |
| Smoking status, n (%) | | | | | | 0.47 |
| Never | 722 (76.7) | 801(79.2) | 779(78.2) | 795(79.1) | 833 (76.7) | |
| Once in a while | 27(2.7) | 18(1.8) | 22(2.2) | 22(2.2) | 17(1.6) | |
| Usually | 207(20.6) | 193(19.1) | 195(19.6) | 188(18.7) | 236 (21.7) | |
| Alcohol status, n (%) | | | | | | 0.20 |
| Never | 826(82.1) | 849(83.9) | 842(84.5) | 864(86.0) | 919 (84.7) | |
| Once in a while | 149(14.8) | 136(13.4) | 136(13.7) | 122(12.1) | 143(13.2) | |
| Usually | 31(3.1) | 27(2.7) | 18(1.8) | 19(1.9) | 24(2.2) | |
| Physical activity level, n (%) | | | | | | 0.002 |
| Low | 538(53.5) | 566(55.9) | 534(53.6) | 573(57.0) | 651(59.9) | |
| Medium | 435(43.2) | 421(41.6) | 447(44.9) | 396(39.4) | 407(37.5) | |
| High | 33(3.3) | 25(2.5) | 15(1.5) | 36(3.6) | 28(2.6) | |
| Glucose (mmol/L), mean (SD) | 4.98(1.0) | 5.03(1.1) | 4.99(1.0) | 5.00(1.0) | 5.57(3.0) | <0.001 |

CI: 2.38, 4.50; $p < 0.001$), compared to individuals in the lowest DII quintile (Table 3, Fig. 2).

The results of the stratified and interaction analysis are shown in Fig. 3 and Table S4. The correlation between DII and T2DM was stronger among subjects with low physical activity level ($OR_{Q5 vs Q1}$: 5.62, 95% CI: 3.00, 10.51), obese subjects ($OR_{Q5 vs Q1}$: 4.62, 95%

**Table 3  Results of multivariate logistic regression models examining the relation between the Dietary Inflammatory Index and T2DM.**

| | No. T2DM (%) | Unadjusted | | Sex- and ethnicity- adjusted | | Fully adjusted | |
|---|---|---|---|---|---|---|---|
| | | OR(95% CI) | *p* | OR(95% CI) | *p* | OR(95% CI) | *p* |
| DII as continuous variable per 1-piont increase | 419 | 1.05(1.02,1.08) | 0.001 | 1.06(1.03,1.08) | <0.001 | 1.05(1.02,1.08) | <0.001 |
| DII as categorical variable | | | | | | | |
| Q1 | 60 | 1(reference) | | 1(reference) | | 1(reference) | |
| Q2 | 63 | 1.05(0.73,1.51) | 0.81 | 1.12(0.77,1.62) | 0.55 | 1.12(0.77,1.62) | 0.55 |
| Q3 | 65 | 1.10(0.77,1.58) | 0.60 | 1.17(0.82,1.69) | 0.39 | 1.14(0.79,1.65) | 0.48 |
| Q4 | 66 | 1.11(0.77,1.59) | 0.58 | 1.18(0.82,1.70) | 0.37 | 1.15(0.80,1.66) | 0.46 |
| Q5 | 165 | 2.83(2.07,3.85) | <0.001 | 3.37(2.46,4.63) | <0.001 | 3.27(2.38,4.50) | <0.001 |

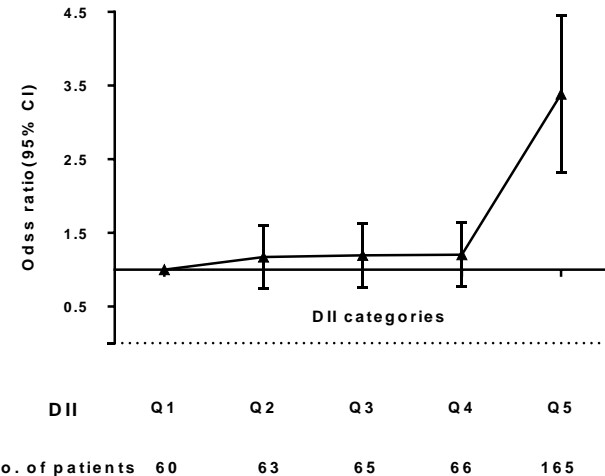

**Figure 2  Odds ratio (95% CI) of Dietary Inflammatory Index (DII) and T2DM.**

CI: 2.50, 8.56), Uyghurs (OR$_{Q5}$ vs$_{Q1}$: 4.11, 95% CI: 1.85, 9.14) and participants ≥ 55 years (OR$_{Q5}$ vs$_{Q1}$: 5.14, 95% CI: 2.79, 9.47).

Additionally, we conducted a sensitivity analysis for unmeasured confounding in this study using E-values (*Linden, Mathur & VanderWeele, 2020*). E-values can be used to estimate the minimum correlation strength associated with outcome indicators for unmeasured confounders that can explain away the research results, so as to evaluate the robustness of the association between exposure factors and outcome indicators, and when the *E*-value is relatively large, it may provide strong evidence to support the relationship. In our study, the *E* value of the point estimate was 5.99 that is higher than the observed risk ratio 3.27, so it provided the evidence to support the relationship between DII and T2DM (Fig. S4).

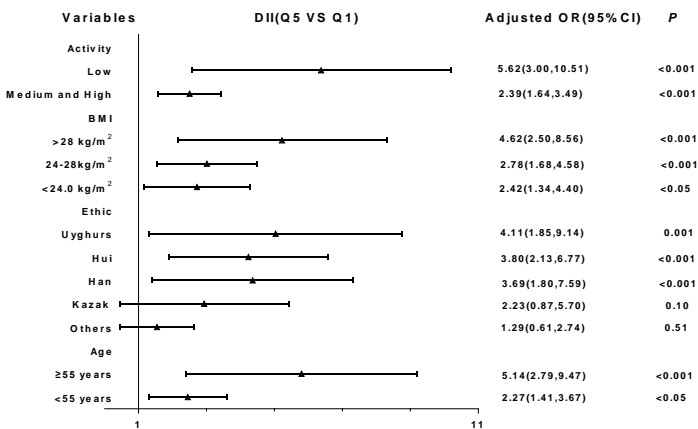

| Variables | DII(Q5 VS Q1) | Adjusted OR(95%CI) | P |
|---|---|---|---|
| Activity | | | |
| Low | | 5.62(3.00,10.51) | <0.001 |
| Medium and High | | 2.39(1.64,3.49) | <0.001 |
| BMI | | | |
| >28 kg/m² | | 4.62(2.50,8.56) | <0.001 |
| 24-28kg/m² | | 2.78(1.68,4.58) | <0.001 |
| <24.0 kg/m² | | 2.42(1.34,4.40) | <0.05 |
| Ethic | | | |
| Uyghurs | | 4.11(1.85,9.14) | 0.001 |
| Hui | | 3.80(2.13,6.77) | <0.001 |
| Han | | 3.69(1.80,7.59) | <0.001 |
| Kazak | | 2.23(0.87,5.70) | 0.10 |
| Others | | 1.29(0.61,2.74) | 0.51 |
| Age | | | |
| ≥55 years | | 5.14(2.79,9.47) | <0.001 |
| <55 years | | 2.27(1.41,3.67) | <0.05 |

**Figure 3** Stratified analysis of the association between DII and T2DM after adjusting for potential confounding factors (Quartile 5 vs. Quartile 1).

## DISCUSSION

In the baseline survey of a Xinjiang multi-ethnic cohort study, we found an positive correlation between DII and T2DM. Those with the most pro-inflammatory diets had a higher risk of developing T2DM compared to participants with the lowest quintile (most anti-inflammatory diets). This increased risk related to pro-inflammatory diet was independent of other diabetes risk factors. This finding is in accord with prior publications from the Mexico City Diabetes Mellitus Survey (*Denova-Gutierrez et al., 2018*), where subjects in the highest DII quintile were observed to be about three times more likely to develop T2DM than those in the lowest DII quintile.

In this study, we used the previously derived and validated DII scores to assess the effect of a pro-inflammatory diet on T2DM. The DII score for The Xinjiang population included 27 food parameters, with a mean of 0.81. The mean of DII in our study was slightly higher than in other studies. For example, the mean value of DII in a large French prospective cohort of women was −0.06 (*Laouali et al., 2019*) and a study in a Japanese population reported a mean of DII as −2.07 (*Kotemori et al., 2020*), both of which are lower than what we observed in our population. This may result from ethnic and culinary differences among participants that were analyzed in our study. Our objects were from Yili prefecture in Xinjiang, which belongs to the area inhabited by ethnic minorities. The diet in this area is mainly composed of pasta, beef, mutton, cream, and other of pro-inflammatory diet components. The intake of vegetables, fruits, and other anti-inflammatory diet components is relatively low in this area. This is consistent with previous reports that differences in nutrient intake or food consumption are associated with DII (*Alipoor et al., 2019*; *Ashton et al., 2020*). Lower C-reactive protein concentration was associated with higher intake of fruits and vegetables (*Shivappa et al., 2019*), legumes, nuts, and low-fat dairy products (*Ferreira et al., 2019*; *Lohman et al., 2019*). Previous studies have also reported associations between intake of certain nutrients, such as total dietary fiber intake, moderate alcohol

consumption (*Padin et al., 2019*), vitamin E and vitamin C intake (*Oliveira et al., 2019*), and lower levels of inflammatory markers (*Hebert et al., 2014*).

This research has shown that DII is related to an increased risk of T2DM, a non-communicable chronic disease of great public health importance. Higher dietary inflammatory potential increases the risk of obesity and its complications, such as T2DM and cardiovascular disease (*Zhong et al., 2017*). In our study, we observed that subjects in the highest DII quintile were at greater risk for developing T2DM than subjects in the lowest DII quintile. Our analysis was consistent with previous studies that have assessed the relationship between diet and T2DM. One study among middle-aged South African women indicated that DII is associated with T2DM risk through obesity, particularly central (i.e., abdominal) obesity (*Mtintsilana et al., 2019*). Studies in pre-diabetes have also found that an anti-inflammatory diet prevents the progression of pre-diabetes into frank diabetes (*Jacobo-Cejudo et al., 2017*). Multiple intervention studies have shown that an anti-inflammatory diet high in carbohydrates, low in fat, and relatively high in fiber reduces the incidence of T2DM by 50%, possibly because a high-fat diet increases inflammatory cytokines, leading to insulin resistance and high blood sugar (*McGeoch et al., 2013*).

The link between DII and T2DM can be explained by the effect of a pro-inflammatory diet on insulin resistance, which is associated with the inflammatory state (*Roden & Shulman, 2019*). Previous studies have shown that systemic, chronic low-grade inflammation characterized by continuous elevation of inflammatory factors in the circulatory system is associated with the occurrence and development of T2DM (*Gardener, Rainey-Smith & Martins, 2016*; *Nilholm et al., 2018*). Inflammatory factors such as IL-6 (*Fadaei et al., 2020*), TNF-$\alpha$, and CRP have been shown to predict the risk of T2DM (*Akour et al., 2018*); however, the specific mechanism remains unclear (*Naidoo, Naidoo & Ghai, 2018*). Diet may affect the development of T2DM by changing the body's inflammatory state. However, prospective analysis is required to test whether this occurs in the natural history of the disease.

Our study had considerable strengths. First, to our knowledge, this work was first to study the potential of diet to promote inflammation and T2DM in Xinjiang. Second, our study was conducted in the Xinjiang Uyghur Autonomous Region, located in the northwest of China, which is associated with considerable ethnic and culinary heterogeneity, which could broaden the distribution of DII thus making it easier to observe a true DII-T2DM relationship. This study involved baseline surveys of more than 31,000 subjects in the three regions of Xinjiang and provided a stable field support for the current study. Third, our participants were from multi-ethnic populations, and therefore the association between DII and T2DM in our study was able to account for potentially confounding factors related to ethnicity. Thus, we were able to show that the DII can be applied to diverse populations as described in other studies (*Namazi, Larijani & Azadbakht, 2018*; *Ruiz-Canela et al., 2015*); and this, in turn, enables comparison with population-based findings in many different regions of the world.

Our study also had some limitations. We used a cross-sectional design, our analysis relied on observational data, and there is a chance that our findings could be explained

directly by causation. It was not clear whether T2DM patients were more likely to choose a pro-inflammatory diet or whether a pro-inflammatory diet helped promote or maintain T2DM. Future cohort studies are required to validate the association between T2DM and DII. As with most observational studies, the FFQs in this study were self-administered; therefore, dietary intake may the associated with reporting biases. In order to minimize such problems, we designed the questionnaire according to local dietary conditions and used food mold during the study.

## CONCLUSIONS

In summary, our study suggests that a diet with a pro-inflammatory potential is associated with an increased risk of T2DM. This study included participants from the baseline study in the Xinjiang multi-ethnic natural population cohort construction and health follow-up study. Our results add to the understanding of the mechanisms of diet-related inflammation and T2DM. Moreover, DII may be an important tool for characterizing inflammatory diets. Finally, further longitudinal surveys are needed to assess the relationship and determine cause and effect.

## ACKNOWLEDGEMENTS

The authors are indebted to all participants for their continued participation. They are also grateful to all members of the Xinjiang multi-ethnic cohort study group.

### Funding

This work was supported by the provincial and ministerial joint project of the State Key Laboratory for the Prevention and Treatment of high morbidity in Central Asia (NO. SKL-HIDCA-2019-), the National key research and development plan "precise medical research" key special sub-project "Xinjiang multi-ethnic natural population cohort construction and health follow-up study" (NO. 2017YFC0907203) and Xinjiang Uygur Autonomous Region "13th Five-Year" Key Discipline (Plateau discipline)-Public Health and Preventive Medicine. The funders had no role in study design, data collection and analysis, decision to publish, or preparation of the manuscript.

### Grant Disclosures

The following grant information was disclosed by the authors:
Provincial and ministerial joint project of the State Key Laboratory for the Prevention and Treatment of high morbidity in Central Asia: SKL-HIDCA-2019-.
National key research and development plan "precise medical research" key special sub-project "Xinjiang multi-ethnic natural population cohort construction and health follow-up study": 2017YFC0907203.
Autonomous Region "13th Five-Year" Key Discipline (Plateau discipline)-Public Health and Preventive Medicine.

## Competing Interests

The authors declare there are no competing interests. Dr. James R. Hébert owns controlling interest in Connecting Health Innovations LLC (CHI), a company that has licensed the right to his invention of the Dietary Inflammatory Index (DII®) from the University of South Carolina in order to develop computer and smart phone applications for patient counseling and dietary intervention in clinical settings. Dr. Nitin Shivappa is an employee of CHI.

## Author Contributions

- WenHui Fu conceived and designed the experiments, performed the experiments, analyzed the data, prepared figures and/or tables, authored or reviewed drafts of the paper, and approved the final draft.
- Hualian Pei performed the experiments, prepared figures and/or tables, and approved the final draft.
- Nitin Shivappa and James R. Hebert conceived and designed the experiments, authored or reviewed drafts of the paper, supervision, and approved the final draft.
- Tao Luo performed the experiments, analyzed the data, prepared figures and/or tables, data curation, and approved the final draft.
- Tian Tian and Zewen Zhang performed the experiments, analyzed the data, prepared figures and/or tables, and approved the final draft.
- Dilibaier Alimu performed the experiments, prepared figures and/or tables, data curation, and approved the final draft.
- Jianghong Dai conceived and designed the experiments, authored or reviewed drafts of the paper, supervision, project administration, funding acquisition, and approved the final draft.

## Human Ethics

The following information was supplied relating to ethical approvals (i.e., approving body and any reference numbers):

The study was conducted in accordance with the Declaration of Helsinki, and the protocol was approved by the Ethics Committee of Ethics Committee of Xinjiang Uygur Autonomous Region Academy of Traditional Chinese Medicine (2018XE0108).

## Data Availability

Raw data is available in the Supplemental Files.

## Supplemental Information

Supplemental information for this article can be found online at http://dx.doi.org/10.7717/peerj.11159#supplemental-information.

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
