# Peer review of "Association between Dietary Inflammatory Index and Type 2 diabetes mellitus in Xinjiang Uyghur autonomous region, China"

_PeerJ, doi:10.7717/peerj.11159_

## Round 0.1 · original submission · Major Revisions

While both reviewers felt this study had merit, reviewer-2 in particular has raised issues with the analysis, and each of these points should be carefully addressed in a revision.

The issues cited under Basic Reporting are largely matters of presentation of data: these should be straightforward to address. Similarly the comments under experimental analysis reflect need for clarification. Although extensive, these issues should not present too much of a problem for you.

Under 'Validity of findings' though a few more detailed issues arise. The use of one-way ANOVA is not appropriate and needs to be changed, and statistical analyses require reporting of the p value when the word 'significant' is used. Point 4 in this section requires clarification and re-formatting to display the 95% CIs and clear statements about whether the odds ratios are significant. Points 6 and 7 are matters of clarification.

Please address all of these issues should you decide to resubmit, and detail your changes in a cover letter.

Note that this may need to be returned to reviewer-2 for further comment.

·

Basic reporting

No comment

Experimental design

In relation to methods:
I suggest the authors include in the section: Dietary Assessment (line 67) references of the validated FFQ and about Physical Activity (line 121).

Validity of the findings

The authors applied appropriate statistical test.
The data are robust and the conclusion support the originate question investigated.
The limitations of the study were clearly pointed out in the text.
But in the section Conclusion: line 242 the authors described “...that a diet with a high pro-potential is associated with an increased risk of T2DM.” I suggest include the word inflammatory:
I would recommend something like: “a diet with a pro-inflammatory potential is associated with an increased risk of T2DM...”.

Additional comments

I think this manuscript has interesting findings.

Reviewer 2 ·

Basic reporting

1) Table1: Firstly, rounding digits should be consistent, for example, most of the numbers are rounded to one digit, but for Kazak T2DM group, the percentage is 9.55. Secondly, in the footnote, it is unclear how the continuous variables were compared. For footnote A, if the authors added the comparison method to the Method Section, it’ll be unnesscrary to include this footnote. Footnote B should be removed and just add it right behind the variable name, just like the other variables.
2) Table2: Firstly, the title should be revised to ‘baseline characteristics of study population by Dietary Inflammatory Index(DII) categorization’, Secondly, it should be quintiles instead of quartiles. Thirdly, the cutoff of categories should be listed as ‘<-5.33’,’-5.33— 1.34’,’1.35-1.31’ etc., but not the exact number alone. Fourthly, for glucose, are these mean and standard error?
3) Figure1: Title should be revised, also there is no need to list the categories of each covariate. In addition, the exact numbers of odds ratio and its 95% CI isn’t listed throughout the paper.
4) Figure2: Instead of having a plot and a table pasted together, this figure can be revised as a forest plot: showing the odds ratio of the outcome with T2DM vs no T2DM for patients in vs. out of the respective subgroups. (e.g. show the variables and categories in the left column, and then another column to show the number of OR(95% CI), the last column would be the plot. Again no need to list the categories of each covariate.
5) Figure 3: Can be listed as a supplemental figure, but not a main figure.

Experimental design

1) In the statistical method section, please specify which comparisons tests were used in table1. In addition, the authors performed parametric tests for the comparisons, but it is unclear if the normality assumption was checked for the continuous variables.
2) In the statistical method section, how did the authors decide which variables to include in the multivariable model? Moreover, the authors should list the multivariable adjustments used.
3) The authors should be cautious about stating something is significant. I suggest that authors to revise the statement as p<0.05 was considered to be ‘significantly’ significant’ in line 132. This language should be revised throughout the paper.
4) In the sensitivity analysis part, the authors used two different ways, using imputation and E-values, however, it is unclear how the imputations were performed, and if there are missing values in the raw data, the missing percentage of each variables should be reported.

Validity of the findings

1) In the results section, the authors summarized some findings from table1 and table2. For example, in line 146, the authors stated that ‘subjects with Hui or Uyghur ethnicity had higher prevalence of T2DM’. However, this is not safe to draw this conclusion. Firstly, the authors should add the reference group; Secondly, in comparisons consisting of two or more categorical independent variables, such as age group, BMI, ethnicity, education level, etc., a one-way ANOVA would not be suitable to draw this conclusion, post hoc multiple comparison tests should be performed to acquire correct p-values.
2) In line 144, the authors mentioned the age of the participants ranged from 45-60 years old, however in Table1, it did show the study has included patients <45 years old. This should be revised to be consistent.
3) When stating the findings is statistically significant or evident, if the tests were performed, the corresponding p-values should be included.
4) In Line 155, the authors reported the odds ratios for all of the groups, but only included the 95% CI for the 5th quintile. It is unclear if the other odds ratios are statistically significant; and in line 157, the authors only reported the odds ratio for the 5th quintile. It would be better if the authors could show these as a table or figure, instead of listing it in the text.
5) In line 156, ‘co-variables’ should be revised to ‘covariates’. In line 157, ‘over 50% greater odds of having’ is not accurate.
6) It is not clear how the stratified analysis was done.
7) The interpretation about E-values is not clear enough.

Additional comments

This study discussed the association between pro-inflammatory diet and having T2DM in Xinjiang district of China, which provide valuable information for further studies. However, it still requires revisions.

---

## Round 0.2 · Minor Revisions

We are nearly there; please could you attend to the minor revisions noted by the reviewer and resubmit a suitably revised paper.

This will not need to go to further review.

Reviewer 2 ·

Basic reporting

no comment

Experimental design

1. In table1, 'referent' should be revised as 'reference'.
2. In table2, the statistics should be specified for glucose, eg., 'glucose(mmol/L), mean(SD)'.
3. In statistical analysis section, line 139, 'Values were determined using a student’s t-test for continuous variables and Chi-square test for categorical variables', where 'values' should be revised as 'P-values'. In addition, the authors should also add a sentence stating the normality was checked using xxx test.
4. It was not clear how the stratified analyses were performed, was the comparison being made between model with/without interaction terms?
5. In the sensitivity analysis section, the authors stated that multiple imputation was performed, however, the authors failed to stating the results for the model after imputation. Were there any differences between the models before and after imputation?

Validity of the findings

no comment

Additional comments

Thank you for addressing my questions.

---

## Round 0.3 · accepted · Accept

Thanks for these final corrections.